# Understanding cauda equina syndrome: protocol for a UK multicentre prospective observational cohort study

Julie Woodfield,[1,2] Ingrid Hoeritzauer,[1,2] Aimun A B Jamjoom,[2,3] Savva Pronin,[2] Nisaharan Srikandarajah,[4] Michael Poon,[1] Holly Roy,[5] Andreas K Demetriades,[1] Philip Sell,[6] Niall Eames,[7] Patrick F X Statham,[1] British Neurosurgical Trainee Research Collaborative (BNTRC)

For numbered affiliations see end of article.

**Correspondence to**
Julie Woodfield;
julie.woodfield@ed.ac.uk

## ABSTRACT

**Introduction** Cauda equina syndrome (CES) is a potentially devastating condition caused by compression of the cauda equina nerve roots. This can result in bowel, bladder and sexual dysfunction plus lower limb weakness, numbness and pain. CES occurs infrequently, but has serious potential morbidity and medicolegal consequences. This study aims to identify and describe the presentation and management of patients with CES in the UK.

**Methods and analysis** Understanding Cauda Equina Syndrome (UCES) is a prospective and collaborative multicentre cohort study of adult patients with confirmed CES managed at specialist spinal centres in the UK. Participants will be identified using neurosurgical and orthopaedic trainee networks to screen referrals to spinal centres. Details of presentation, investigations, management and service usage will be recorded. Both patient-reported and clinician-reported outcome measures will be assessed for 1 year after surgery. This will establish the incidence of CES, current investigation and management practices, and adherence to national standards of care. Outcomes will be stratified by clinical presentation and patient management. Accurate and up to date information about the presentation, management and outcome of patients with CES will inform standards of service design and delivery for this important but infrequent condition.

**Ethics and dissemination** UCES received a favourable ethical opinion from the South East Scotland Research Ethics Committee 02 (Reference: 18/SS/0047; IRAS ID: 233515). All spinal centres managing patients with CES in the UK will be encouraged to participate in UCES. Study results will be published in medical journals and shared with local participating sites.

**Trial registration number** ISRCTN16828522; Pre-results.

## Strengths and limitations of this study

► This UK-wide study will be the largest prospectively established cohort of patients with cauda equina syndrome (CES).

► The collection of detailed clinical data will describe the range of presentations treated as CES in the UK in current practice and allow stratification of findings by clinical presentation.

► Validated outcome measures will be used to assess pain, disability, and bladder, bowel and sexual functions 1 year after treatment.

► Participant identification and recruitment will be efficiently carried out using trainee research networks to identify participants when referred urgently to specialist spinal centres.

► The relationship of timing of investigation and decompression to patient outcome will be limited by patient and clinician reporting of the timing of symptom onset.

## INTRODUCTION

Cauda equina syndrome (CES) is a rare but potentially devastating condition caused by compression of the cauda equina nerve roots. This most commonly occurs due to a prolapsed intervertebral disc. The clinical syndrome includes any of bilateral sciatica, saddle anaesthesia, bladder, bowel or sexual dysfunction.[1–3] The disabling nature of these symptoms causes significant medical and social morbidity and high healthcare and social-care costs. In addition, litigation related to the management of CES leads to significant medicolegal workload and costs.[1 4 5]

Due to the consequences of CES for patients and society, several groups have issued clinical guidance on standards of care for CES.[1 6–8] However, the evidence base for current clinical guidance consists of small retrospective single-centre case series.[1 9 10] Even systematic reviews of outcomes in CES have included relatively few patients, with the largest including 464 patients.[9 11] Lack of a clear definition of CES has hampered comparative analysis of historical studies, and different interpretations of available evidence have been offered.[10 12] A diagnosis of CES encompasses patients presenting with mild to severe urinary and bowel symptoms, perineal or perianal numbness, sexual function

disturbance or bilateral sciatica, and patients may also experience lower limb weakness, numbness or unilateral sciatica.[2 3 13] Outcomes for different presentations vary and accurate division by presentation may help to clarify the understanding of outcome studies and develop care standards appropriate to the presentation.[1 14]

Retrospective case series in the UK have identified approximately 15–31 patients per year per specialist neurosurgical or spinal centre with confirmed CES.[3 13 15 16] Published estimates of the incidence of CES are fewer than one case per 100 000 population.[17 18] However, in 2010–2011 in England, 981 surgical decompressions were performed for CES,[19] and the population was estimated at 52 234 000[20] giving an incidence of 1.9 per 100 000. Therefore, there may be over 1000 patients managed for CES in the UK each year. Accurate data on the presentation and management of these patients would establish current management plus adherence to and feasibility of care-quality statements, as well as potentially informing the revision of guidance based on accurate and current data.

The British Neurosurgical Trainee Research Collaborative (BNTRC) has previously successfully used a network of neurosurgical trainees across the UK and Ireland to identify cases via local tertiary referral systems in conditions such as chronic subdural haematoma.[21] As CES is managed in the UK by specialist spinal services, similar case ascertainment via specialist referral systems to neurosurgical, orthopaedic or joint spinal services provides a method of accurately identifying patients with CES during hospital admission. We propose to carry out the first national cohort study of the presentation and management of CES in the UK and establish the largest prospective series of patients with CES. This will provide data on CES incidence, epidemiology, presentation, management and outcomes. This will inform the development of clinical guidance and identify areas for future research in CES.

This prospective observational cohort study aims to:
- ► Identify the number of cases of CES in the UK in all collaborating centres.
- ► Describe the presenting symptoms and signs in patients with CES.
- ► Describe the pathways of presentation to specialist spinal services for patients with CES in the UK.
- ► Describe the type, timing and findings of investigations in patients with CES.
- ► Describe the medical and surgical management of CES.
- ► Compare current practice to standards of care for CES.
- ► Describe clinical outcomes for patients with CES using validated patient reported outcome measures stratified by presentation, investigation findings and management.
- ► Demonstrate the ability of neurosurgical and orthopaedic surgical trainee networks to collaborate successfully on a prospective cohort study.

## METHODS AND ANALYSIS

Understanding Cauda Equina Syndrome (UCES) is a prospective cohort study of patients with confirmed CES managed at specialist spinal centres in the UK. Cases will be identified by neurosurgical or orthopaedic trainees in each specialist centre through daily screening of tertiary referrals and admissions to specialist spinal services. All patients managed as CES by the treating team will be included in this study.

Data regarding timing and type of symptom onset, referral, investigation, management and outcome will be recorded anonymously on a secure database by the local trainee investigator during the patient's hospital admission and after discharge. Patients' consent will be sought for the use of their data and they will be asked to complete patient-reported outcome measures representing their condition before surgery and up to 1 year after surgery. Imaging at presentation will also be collected. This data will be compared with care quality statements and published outcome data for CES. This is an observational study. No changes to routine patient care will occur during this study.

### Participant selection
The study will recruit patients for 1 year. Cases will be identified from admissions to spinal units between 1 June 2018 and 31 May 2019. The last 1-year follow-up assessments will be sent to participants on 31 May 2020.

For inclusion in this study, the patient must:
- ► Be over 18 years old.
- ► Be admitted to a specialist spinal service in the UK between 1 June 2018 and 31 May 2019.
- ► Have capacity to provide informed consent for participation in this study.
- ► Have a diagnosis of clinical CES and structural compression of the cauda equina on imaging as determined by the treating clinician.
  - – Clinical CES includes any of : altered saddle sensation; bladder dysfunction; bowel dysfunction; sexual dysfunction; or bilateral sciatica. This should be associated with radiological compression of the cauda equina. The cauda equina compression can be due to any cause, including but not limited to disc, tumour or infection.

There is no upper age limit as we aim to establish the demographics of those presenting with CES.

The exclusion criteria are:
- ► Children under 18 years of age.
- ► Patients undergoing emergent decompression for unilateral motor or sensory symptoms (such as foot drop), without clinical evidence of CES.
- ► Patients referred with suspected CES where the diagnosis is not confirmed;for example, patients with clinical symptoms and signs of CES but without radiological evidence of cauda equina compression.
- ► Patients not admitted to participating spinal centres in the UK.

- ► Patients admitted to a participating spinal centre before 1 June 2018 or after 31 May 2019.
- ► Patients who are unable to provide informed consent for participation in this study.

Capture-recapture methods will be used to ensure complete case ascertainment. In December 2018, June 2019 and December 2019 all local investigators will check their case ascertainment by asking their local coding departments for all discharges coded as CES using the diagnostic code from the International Classification of Diseases, 10th Revision (ICD-10) G83.4 Cauda Equina Syndrome. Any additional patients identified through this method that meet the inclusion criteria will be invited to participate.

### Data collection

Data relating to presentation, hospital admission, investigations and follow-up will be collected by the local trainee investigator. Data will be collected from the patient's notes, through routine interaction with the patient as part of clinical care, and through interaction with other staff members caring for the patient. All clinical and demographic information collected for this study by the local investigators will be collected routinely. No extra assessments will be performed.

Study participants who have consented to participate will also be asked to fill out details about their patient journey, their symptoms, patient reported outcome measures and service usage. These will be collected electronically and anonymously via the electronic database and linked to the patient record. Patient reported outcome measures will include visual analogue scores for back and leg pain plus relevant sections of the Oswestry Disability Index,[22] the neurogenic bowel dysfunction score,[23] the short form incontinence questionnaire[24] and the Arizona sexual experiences scale.[25]

All patients who are eligible for inclusion in the study will have basic anonymous clinical data collected as part of the screening log to establish participation rates and incidence at each centre. This will allow accurate assessment of the incidence of CES. Patients who do not wish to participate in the study will not be contacted further for the completion of patient reported outcome measures.

The timing and type of clinician reported and patient reported data that will be collected for UCES is shown in figure 1:Study flow diagram.

Clinician-entered data will be entered directly into the database using the participant's unique study number. Imaging will be reviewed on local *p*icture *a*rchive and *c*ommunication *s*ystems (PACS) and transferred to the study team for review. Participant questionnaires will be sent out by email using unique links for each participant. If participants do not have an email address or prefer to fill out questionnaires on paper, paper or telephone versions of the questionnaires will be used. If participants do not respond to the email invitations, they will be contacted to find out whether they wish to continue with the study and to complete the questionnaires. Where

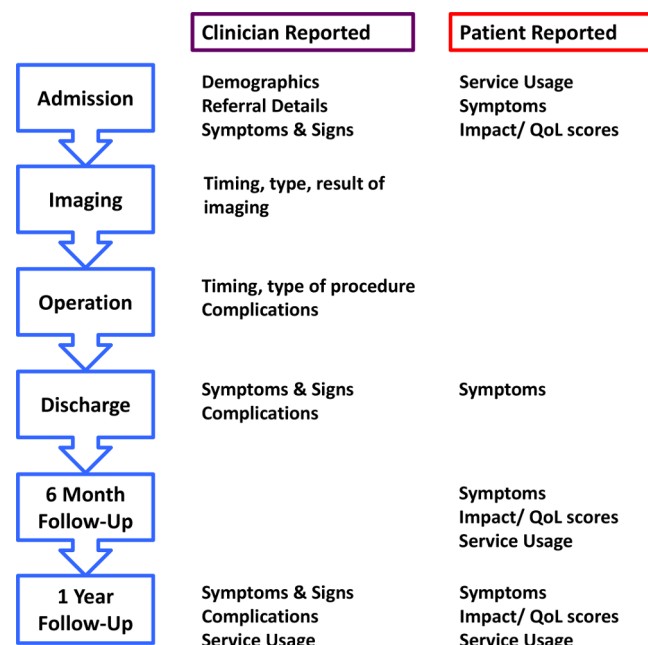

**Figure 1** Study flow diagram. Time points for patient reported and clinician reported data collection in Understanding Cauda Equina Syndrome. (QoL: Quality of Life)

patient data is routinely entered into spinal databases, surgical and outcome data from those databases will be linked anonymously to the patient record by the clinical team using the patient's unique identifier for that database or registry.

### Data analysis

This study aims to establish the number of patients presenting with CES in the UK over 1 year. We expect approximately 20 patients per spinal centre per year depending on the population served and a total of approximately 600–1000 patients in 1 year across the UK. The incidence of CES will be established based on the number of patients identified at each unit and the catchment population of that unit. If all units in the UK participate, incidence will be calculated based on the UK population estimates. Incidence will be calculated from all patients identified as being eligible for the study from referral screening and local coding departments, even if they do not consent for further participation.

A descriptive analysis of the clinical and demographic characteristics of presenting symptoms, signs and outcomes of patients with CES will be performed. This will be determined from both clinician-reported and patient-reported data. CES incidence and characteristics will be broken down into categories such as suspected CES, incomplete CES, with retention CES and early CES, based on the clinical data. The categorical and quantitative findings on imaging will also be described. Methods of patient presentation to specialist services will be described. Type, timings and findings of investigations in patients presenting via different routes will be compared. The investigation and management of

patients with CES will be described and compared with that laid out in current care quality standards. Proportions meeting the standards will be reported. Patient outcomes will be assessed and analysed using both clinician-reported and patient-reported outcome measures at 6 months and 1 year. Patient outcomes will be stratified by demographics, presenting features, causative pathology, timing and findings of investigations, and timing and type of surgery. Patient usage of healthcare services over the year following diagnosis and management of CES, will be assessed using both patient-reported service usage and electronic records.

### Patient and public involvement

The design and aims of this study were discussed with current patients being investigated for CES, and those who had previously been treated for CES. Patients trialled the questionnaires and provided feedback on the questionnaires and patient information leaflet. The length and content of the questionnaires and information leaflet were altered in response to patient feedback. All participants will receive a summary of the results of this study. Patients are not involved in recruitment to this study as this occurs during or after emergency admission to hospital with CES.

## ETHICS AND DISSEMINATION
### Patient consent

Once patients are identified as being eligible to participate in the study, they will be asked by a member of their clinical team whether they would be willing to receive further information about the study. For the majority of patients this will occur during their admission to the spinal unit, and the approach will be made by a member of ward medical or nursing staff. Once verbal consent is gained to give further information about the study, patients will be provided with the information leaflet for the study. Patients who indicate that they are happy to have further discussions regarding the study will be visited in hospital by a member of their clinical team to complete the written consent process. The person undertaking written consent will be adequately trained to do so, and have a good knowledge of the study protocol, aims and processes. The participant will be informed about and consent to their medical records being inspected by regulatory authorities and representatives of the sponsor. Both the participant and the person undertaking the consent will sign and date the informed consent form to confirm that consent has been obtained. The participant will receive a copy of this document and a copy will be filed in the Investigator Site File.

Decompression surgery for CES takes place as an emergency and admissions occur at all times of day and night, throughout the week and weekends. Following decompression, the length of stay in hospital wards may be as short as 1–2 days, or may be longer than a week when there are ongoing bladder or bowel problems. All patients will be given adequate time to read the information leaflet with a minimum time period of 6 hours. Some patients will be discharged prior to being identified as being eligible for the study. These patients will be contacted by telephone by a member of the clinical team and asked if they would be willing to receive information about the study by post or email. If they agree, the information leaflet and consent form will be sent to them, and they will be re-contacted to go through the consent process over the telephone at least 24 hours after receiving the information.

When participants prefer to fill out paper questionnaires or do not respond to the email link, their contact details (name, address and telephone number) will be passed to the central study team at NHS Lothian using the NHS email system with the consent of the patient. The central study team will contact the participants to find out whether they still wish to take part in the study. Those who wish to continue with the study will be sent the questionnaires by email, by post or they can be completed over the telephone with a member of the central study team depending on the preference of the participant. If participants do not wish to continue with the study, they will not be contacted further.

Participants are free to withdraw from the study at any point. If withdrawal occurs, the primary reason for withdrawal will be documented in the participant's electronic case report form. The participant will not be contacted any further for outcome measures but their basic anonymous clinical details will be retained to allow accurate epidemiological assessment of the incidence of CES. If a patient loses capacity to consent for ongoing participation during the course of the study, the data they have already submitted or has already been submitted by their clinical team with their consent will continue to be used in the study, but they will not be contacted with further questionnaires.

### Data protection

All Investigators and study site staff involved with this study will comply with the requirements of the Data Protection Act 1998 with regard to the collection, storage, processing and disclosure of personal information and will uphold the Act's core principles. Access to collated participant data will be restricted to individuals from the research team treating the participants, representatives of the sponsor and representatives of regulatory authorities. Computers used to collate the data will have limited-access measures via user names and passwords. Published results will not contain any personal data that could allow identification of individual participants.

All clinical details will be entered into a database hosted by Castor EDC. Castor EDC complies with all applicable laws and regulations: Good Clinical Practice (GCP), European Union Annex 11 and the European Data Protection Directive. Clinician-entered data will be entered directly into the database using the participant's unique study number. The clinical team can only view the records of

patients from their own centre. Once participants have consented for their email addresses to be stored, these will be entered into the Castor database by the local clinical team. The email address field is stored securely and is encrypted and cannot be viewed by anyone outside of the patient's local centre.

All local investigators will store a copy of the link between the patient's unique study number and their contact details, National Health Service (NHS) number, hospital number, Community Health Index number, unique identifiers for spinal databases or registries or other identifying details on a secure password protected NHS computer. Consent forms and paper completed questionnaires will be stored securely in a locked NHS office. No identifying information will be entered into the secure database except the email address.

All identifiable scans will be stored and transferred within the NHS PACS network. Only anonymised scans will be processed outside the NHS PACS network. Anonymised imaging data will be labelled only with the study number and stored on anonymised CDs or on encrypted hard drives.

### Data retention
All study documentation will be kept for a minimum of 5 years from the end of the study. When the minimum retention period has elapsed, study documentation will not be destroyed without permission from the sponsor. The end of the study is 18 months after the enrolment of the last participant.

### Insurance and indemnity
Sites participating in the study will be liable for clinical negligence and other negligent harm to individuals taking part in the study and covered by the duty of care owed to them by the sites concerned. The sponsor requires individual sites participating in the study to arrange for their own insurance or indemnity in respect of these liabilities. Sites which are part of the UK NHS will have the benefit of NHS Indemnity.

### Ethical review
The study will be conducted in accordance with the principles of the International Conference on Harmonisation Tripartite Guideline for GCP. All researchers are encouraged to undertake GCP training in order to understand the principles of GCP. However, this is not a mandatory requirement. GCP training status for all investigators should be indicated in their respective curriculum vitaes.

Local management approvals must be in place at each site prior to recruitment of patients to this study. The most recent version of the protocol will be available on the website of the BNTRC at www.bntrc.org.uk. This study is sponsored by NHS Lothian.

### Peer review
The concept for this study was selected by a panel of judges in an open competition for support from the BNTRC. The protocol has been reviewed and approved by the steering committee for this study and reviewed by the British Orthopaedic Trainees' Association, the British Association of Spine Surgeons, and the BNTRC committee.

### Publication
Ownership of the complete dataset arising from this study resides with the steering committee and the BNTRC. On completion of the study, the data will be analysed, tabulated and a report will be prepared. A summary report of the study will be provided to the REC within 1 year of the end of the study. Local data collected as part of this study belongs to the local team collecting that data. The study report will be used for publication and presentation at scientific meetings. Summaries of results will also be made available to local investigators. Following the initial analysis and publication, study data will be made available to those who submit successful peer-reviewed proposals for use of the data to the steering committee via the BNTRC.

All local investigators who enter data for at least one case will be named as contributors in publications arising from this study and will receive a certificate of collaboration in this study. Authorship of publications arising from this study will be determined in accordance with the guidelines of the International Committee of Medical Journal Editors.[26]

**Author affiliations**
[1]Department for Clinical Neurosciences, Western General Hospital, Edinburgh, UK
[2]Centre for Clinical Brain Sciences, The University of Edinburgh, Edinburgh, UK
[3]Department of Neurosurgery, Aberdeen Royal Infirmary, Aberdeen, UK
[4]Department of Neurosurgery, Walton Centre NHS Foundation Trust, Liverpool, UK
[5]South West Neurosurgery Centre, Derriford Hospital, Plymouth, UK
[6]Centre for Spinal Studies and Surgery, Queens Medical Centre, Nottingham, UK
[7]Department of Trauma and Orthopaedics, Royal Victoria Hospital, Belfast, UK

**Acknowledgements** The authors would like to acknowledge the guidance and support in developing this protocol from the study steering committee (Simon Clark, AKD, Gareth Dobson, NE, IH, AABJ, Andraay Leung, MP, Manjunath Prasad, SP, PFXS, NS, PS, Stacey Darwish, Martin Wilby, JW), the British Association of Spine Surgeons (Mike Hutton, David Cumming, Jake Timothy, Andraay Leung, Chrishan Thakar, Tim Germon), the British Orthopaedic Trainees Association (Rory Morrison), the British Neurosurgical Trainee Research Collaborative (Ellie Edlman, Ruichong Ma, Michael Poon), and the sponsor NHS Lothian (Kenneth Scott). The authors would also like to thank all patients at the Department of Clinical Neurosciences in Edinburgh who provided input into the design of the study and the questionnaires.

**Collaborators** The British Neurosurgical Trainee Research Collaborative, the British Orthopaedic Trainees Association, the British Association of Spine Surgeons.

**Contributors** JW, IH, SP and AABJ contributed to the conception of the study, design of the study, writing of the protocol, revision of the protocol and approving the final version. NS, MP, HR, AKD, PS, NE, PFXS and the BNTRC contributed to the design of the study, revision of the protocol and approving the final version.

**Funding** The authors have not declared a specific grant for this research from any funding agency in the public, commercial or not-for-profit sectors.

**Competing interests** JW reports grants from the Wellcome Trust during this study. IH reports grants from the Association of British Neurologists/Patrick Berthoud Charitable Trust during this study. AABJ, SP, NS, MP, HR, AKD, PS, NE, PFXS and the BNTRC have nothing to disclose.

**Patient consent** Obtained.

**Ethics approval** South East Scotland Research Ethics Committee 02.

**Provenance and peer review** Not commissioned; externally peer reviewed.

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
