## [Reviewer comments · BMJ Open]

This paper was submitted to a another journal from BMJ but declined for publication following peer review. The authors addressed the reviewers' comments and submitted the revised paper to BMJ Open. The paper was subsequently accepted for publication at BMJ Open.

(This paper received three reviews from its previous journal but only two reviewers agreed to published their review.)

ARTICLE DETAILS

TITLE (PROVISIONAL)	Understanding Cauda Equina Syndrome: protocol for a United Kingdom multi-centre prospective observational cohort study
AUTHORS	Woodfield, Julie; Hoeritzauer, Ingrid; Jamjoom, Aimun A.B.; Pronin, Savva; Srikandarajah, Nisaharan; Poon, Michael; Roy, Holly; Demetriades, Andreas; Sell, Philip; Eames, Niall; Statham, Patrick; Research Collaborative, British Neurosurgical Trainee

VERSION 1 – REVIEW

REVIEWER	N V Todd Newcastle Nuffield Hospital, Clayton Rd, Newcastle upon Tyne
REVIEW RETURNED	20-Jul-2018

GENERAL COMMENTS	The authors might wish to restrict the study only to patients with disc prolapses. This will be the largest group anyway. I do not believe that other pathologies causing CES such as for example spinal stenosis, cancer or infection are directly comparable to CES caused by lumbar disc prolapses. As a minimum the authors should say prospectively that the disc prolapse patients will be analysed separately. The neurogenic bowel score was based upon patients with spinal cord injury and it contains a domain that addresses autonomic dysreflexia which will not be an issue in the CES patients. The score will need to be modified to exclude that domain. The subclassifications of CES could include CESE (CES early) [Todd NV BJNS 1917] which is point in the evolution of CES just prior to CESI. This could be a hugely important study. It will simultaneously be the largest study of CES patients ever and the only prospective study. I strongly support this study.
---

REVIEWER	Sashin Ahuja University Hospital of Wales. UK
REVIEW RETURNED	05-Aug-2018

GENERAL COMMENTS	The authors outline the assessment and scoring of the symptoms on Page8 Line 45-(P8L45). One assumes that the clinical assessment data sheet would include
---

	the clinical examination (eg assessment of perianal sensations and digital per rectal examination etc) and maybe post void bladder scan . Are the authors hoping to assess the specificity and sensitivity of these clinical signs taking into account the power of the study with the total number of patients that they are hoping to recruit.
--	---

VERSION 1 – AUTHOR RESPONSE

Reviewer: 1

Reviewer Name: N V Todd

Institution and Country: Newcastle Nuffield Hospital, Clayton Rd, Newcastle upon Tyne

Please state any competing interests or state ‘None declared’: None declared.

Please leave your comments for the authors below

The authors might wish to restrict the study only to patients with disc prolapses. This will be the largest group anyway. I do not believe that other pathologies causing CES such as for example spinal stenosis, cancer or infection are directly comparable to CES caused by lumbar disc prolapses. As a minimum the authors should say prospectively that the disc prolapse patients will be analysed separately.

We have added that results will be analysed by causative pathology. We expect most cases to be caused by disc prolapse.

The neurogenic bowel score was based upon patients with spinal cord injury and it contains a domain that addresses autonomic dysreflexia which will not be an issue in the CES patients. The score will need to be modified to exclude that domain.

This will be undertaken during analysis of the results. We have added a statement about using the relevant sections of the outcome measures.

The subclassifications of CES could include CESE (CES early) [Todd NV BJNS 1917] which is point in the evolution of CES just prior to CESI.

We have added early CES as a potential category.

This could be a hugely important study. It will simultaneously be the largest study of CES patients ever and the only prospective study. I strongly support this study.

Thank you very much for the support.

Reviewer: 2

Reviewer Name: Sashin Ahuja

Institution and Country: University Hospital of Wales. UK

Please state any competing interests or state ‘None declared’: None to declare

Please leave your comments for the authors below

The authors outline the assessment and scoring of the symptoms on Page8 Line 45-(P8L45). One assumes that the clinical assessment data sheet would include the clinical examination (eg assessment of perianal sensations and digital per rectal examination etc) and maybe post void bladder scan . Are the authors hoping to assess the specificity and sensitivity of these clinical signs taking into account the power of the study with the total number of patients that they are hoping to recruit.

Thank you for your comments. The data collected includes all of the clinical assessments mentioned. We will be reporting the frequency of all of these findings in the cohort. We cannot assess the sensitivity and specificity of signs and symptoms in predicting radiological cauda equina compression because we are not including those without radiological cauda equina compression in this study. Calculating sensitivity and specificity of clinical signs involves including those with and without the condition of interest. This is a cohort study that only includes those with clinical and radiological cauda equina compression.